# Actin-Depolymerizing Factor Gene Family Analysis Revealed That *CsADF4* Increased the Sensitivity of Sweet Orange to Bacterial Pathogens

**DOI:** 10.3390/plants12173054

**Published:** 2023-08-25

**Authors:** Jing Xu, Suming Dai, Xue Wang, Alessandra Gentile, Zhuo Zhang, Qingxiang Xie, Yajun Su, Dazhi Li, Bing Wang

**Affiliations:** 1College of Plant Protection, Hunan Agricultural University, Changsha 410128, Chinawang20220908@163.com (X.W.);; 2National Citrus Improvement Center, Hunan Agricultural University, Changsha 410128, China; 3College of Horticulture, Hunan Agricultural University, Changsha 410128, China; 4Department of Agriculture and Food Science, University of Catania, 95123 Catania, Italy; 5Hunan Plant Protection Institute, Hunan Academy of Agricultural Science, Changsha 410128, China

**Keywords:** *Citrus sinensis*, ADF, phylogenetic analysis, subcellular localization, citrus canker

## Abstract

The actin-depolymerizing factor (ADF) gene family regulates changes in actin. However, the entire ADF family in the sweet orange *Citrus sinensis* has not been systematically identified, and their expressions in different organs and biotic stress have not been determined. In this study, through phylogenetic analysis of the sweet orange ADF gene family, seven CsADFs were found to be highly conserved and sparsely distributed across the four chromosomes. Analysis of the cis-regulatory elements in the promoter region showed that the *CsADF* gene had the potential to impact the development of sweet oranges under biotic or abiotic stress. Quantitative fluorescence analysis was then performed. Seven *CsADFs* were differentially expressed against the invasion of *Xcc* and *C*Las pathogens. It is worth noting that the expression of *CsADF4* was significantly up-regulated at 4 days post-infection. Subcellular localization results showed that CsADF4 was localized in both the nucleus and the cytoplasm. Overexpression of *CsADF4* enhanced the sensitivity of sweet orange leaves to *Xcc*. These results suggest that *CsADFs* may regulate the interaction of *C. sinensis* and bacterial pathogens, providing a way to further explore the function and mechanisms of ADF in the sweet orange.

## 1. Introduction

The sweet orange (*Citrus sinensis* L. *Osbeck*) is an economically important fruit globally [1]. Citrus bacterial pathogens threaten the yield and quality of citrus. Citrus Huanglongbing is caused by *Candidatus* Liberibacter asiaticus (*C*Las) [2]. It is transmitted by diseased plants and *Diaphorina citri*. It is the most destructive disease in citrus production and has damaged industries in many countries and regions [3]. Citrus canker, caused by *Xanthomonas citri* subsp. *citri* (*Xcc*), is one of the most destructive soil-borne diseases, causing serious yield losses [4]. *Xcc* can infect almost all citrus tissues and organs, causing leaf-drop, withered branches, premature fruit-drop, and a decline in fruit quality. Further development of the bacterial disease can lead to the death of young trees, which seriously endangers the healthy development of the citrus industry. Chemical control remains the primary approach, however, this is not conducive to the ecological environment. Therefore, screening for disease-resistant genes and cultivation of citrus disease-resistant varieties are promising economical and effective long-term solutions for controlling diseases. 

The actin-depolymerizing factor (ADF) gene family regulates actin [5]. The first *ADF* gene was isolated and purified from chicken embryo cells more than 30 years ago. Since then, an increasing number of studies have shown that *ADF* is an extremely important regulatory gene family [6]. Actin-depolymerizing factor is a low-molecular-weight (15–22 kDa) conserved gene family in eukaryotic cells. The *ADF* genes have been identified in several plant genomes. For example, 12 *ADF* genes have been identified in *Oryza sativa* [7], 11 in *Arabidopsis thaliana* [8], 13 in *Triticum aestivum* [9], and 14 in *Populus trichocarpa* [10]. 

As ancient proteins, the quantity and function of ADF proteins have undergone constant changes through evolution [10]. Some conserved features of the ADF family extend from animals to plants, including transcriptional and post-transcriptional regulation. In many plants, ADF has a very short first exon, even containing only “ATG” [11]. The structure of the short first exon and long intron enhances the expression of the *ADF* gene in *Petunia* [12]. During the long process of species and gene evolution, the number of plant ADF family members has exceeded that of animals, indicating that the ADF family has more functional differentiation in plants [13,14]. The ADF family is known for its role in depolymerizing microfilaments in polar cells, such as during root hair elongation and pollen tube germination [15]. For example, it is involved in pollen development and pollen tube growth in *Arabidopsis thaliana* [16], *Triticum aestivum* [17], and *hyperoside* [18]. Studies have shown that a lack of *AtADF9* can lead to early flowering [19]. Expression of *AtADF4* in tobacco can cause morphological changes in the plant [20]. Additionally, the ADF expression pattern appears to be tissue-specific. Nine wheat ADF genes (*TaADF4/5/6/7/8/9/23/24/25*) on chromosomes 2 and 6 were highly expressed in the anthers, but their expression levels in the leaves, root tip meristems, roots, stem rachis, and seeds were very low or undetectable [21]. 

Actin-depolymerizing factor plays a crucial role in the response of plants such as Arabidopsis, wheat, rice, and other species to abiotic and biotic stresses [22,23,24]. For instance, *TaADF4* enhances the resistance of wheat plants to wheat stripe rust (*Puccinia striformis* f. sp. *tritici*, *Pst*) infection by regulating the actin cytoskeleton and actively mediating immune responses. *TaADF7* might affect reactive oxygen species (ROS) accumulation and hypersensitive response (HR) by regulating the actin cytoskeleton, thereby promoting wheat resistance to *Pst* infection, whereas *TaADF3* negatively modulates plant resistance to *Pst* [25,26,27]. Some studies have shown that in Arabidopsis, actin cytoskeleton rearrangement mediated by *AtADF4* may be responsible for the transport of AvrPphB and RPS5 to the plasma membrane, thus coordinating plant resistance to *Pst* [28,29]. A DNA fragment of the ADF gene was identified in the top anthracnose-resistant strawberry individuals, which may be related to plant antifungal activity [30].

Actin-depolymerizing factor proteins play a key role in the regulation of plant defenses with pathogen infection. However, there are few reports on ADF in sweet oranges, particularly regarding plant immunity. In the present study, the sweet orange ADF gene family was identified using the whole-genome sequence of the sweet orange. The expression characteristics of *CsADF* genes were analyzed in the interaction between sweet oranges and citrus bacterial pathogens. Moreover, we found that CsADF4 negatively modulated sweet orange resistance against citrus bacterial pathogens. These results provide a basis for effective engineering strategies to improve sweet orange stress tolerance.

## 2. Results

### 2.1. Genome-Wide Identification and Chromosome Mapping of the ADF Gene in Sweet Orange

Based on a strict two-stage screening process, we identified seven ADF genes in sweet oranges, accounting for approximately 0.02% of the entire sweet orange genome (29,445 sweet orange genes). These *CsADFs* members were named according to their chromosome position order (CsADF1—CsADF7), and different transcripts were distinguished by the postscript a/b/c/d. The open reading frame lengths of the ADF genes ranged from 420 to 453 bp, and the calculated theoretical molecular weight (MWs) of ADF proteins ranged from 13.25 to 19.73 kDa. The CsADF protein isoelectric points (pIs) ranged from 5.29 to 8.89 and a GRAVY value of less than zero reflected hydrophilicity. Based on instability index analysis, CsADF1/2/3/4a/4b/5 proteins exhibited an instability index greater than 40, suggesting potential instability. Moreover, the instability index of the CsADF6a/6b/6c/6d/7a/7b proteins ranged from 13.1 to 34.24, indicating possible stability. Notably, the fat index of CsADF was found to be within the range of 62.19 to 80.52, implying the thermal stability of the CsADF protein (Table 1). 

To determine the chromosomal distribution of the *CsADF* genes, their position information was expressed in sweet orange according to the sweet orange Genome Annotation Project sweet orange genome sequence (SWO v3.0). The results showed that the seven CsADF genes were sparsely distributed on four chromosomes, and the gene density was inconsistent on a single chromosome (Appendix A). Most ADF genes are located on chromosome 5, including *CsADF1*, *CsADF2,* and *CsADF3*. *CsADF5* and *CsADF6a/b/c/d* are located on chromosome 8. *CsADF4a/b* and *CsADF7a/b* are located on chromosomes 6 and 9, respectively (Appendix A). 

### 2.2. Phylogenetic Analysis and Collinear Analysis of CsADF Gene Family

In this study, 43 aligned ADF protein sequences from sweet orange (7 ADF proteins), wheat (11 ADF proteins), Populus (14 ADF proteins), and Arabidopsis (11 ADF proteins) were used to generate phylogenetic trees. The phylogenetic distribution showed that higher plant ADFs can be divided into four groups, and because one group is unique to monocotyledons, sweet orange ADF can only be divided into three groups. To further investigate the relationship between genetic differentiation and gene repetition, we performed a collinear analysis. There were four repeats in the ADF genes of sweet orange, and there were repeats in all seven genes except *CsADF6a/b/c/d* (Figure 1B). 

We estimated the Ka/Ks replacement rate of the two protein-coding genes to assess the direction and intensity of the natural selection pressure. The results showed that the replacement rates of the three pairs of genes were all less than 1, between 0.05 and 0.06 (Appendix A). These results suggest that the *ADF* genes of sweet oranges were purified and selected during evolution. 

### 2.3. Structural and Cis-Acting Regulatory Element Analysis of CsADF Genes

We identified five conserved motifs in CsADFs, designated as motifs 1 to 5, and each CsADF had five conserved motifs, except for CsADF6b/c/d, which had four. All *CsADF* proteins contained ADF domains. CsADF1/2/3/6b/6c/6d/7a/7b contains ADF-gelsolin supe domains, which increase the turnover rate of actin and interact with actin monomers and actin filaments. CsADF4a/4b/5 contains ADF-cofilin-like domains, which enhance the turnover rate of actin, and interact with actin monomers (G-actin) as well as actin filaments (F-actin). Except for *CsADF1/5*, all *CsADF* contain one 5’-UTR. *Cis*-regulatory elements (CREs) are generally composed of non-coding DNA, which is the binding site for transcription factors [31]. We determined the number, location, and type of CERs in the promoters of different *CsADF* genes (Appendix A). The results showed that there were significant differences in CERs among the different genes and that there were two to four tandem repeats in some genes, some of which overlapped. 

### 2.4. Expression Profile of CsADF Genes in Different Tissues of Sweet Orange

To evaluate the potential function of the *CsADF* gene in the development of sweet oranges, we analyzed the expression profiles of all seven *CsADF* genes in three sweet orange tissues (stem, leaf, and fruit) using qRT-PCR (Figure 2). Except for *CsADF3*, the expression of the other genes in the leaves was low. Among these, *CsADF2/4a/5/6a* were highly expressed in fruits but poorly expressed in other tissues. It is worth mentioning that the expression of *CsADF6a* in fruits was approximately triple as high as that in other tissues, which suggests *CsADF6a* may play an important role in fruit. *CsADF1* and *CsADF7a* were similarly expressed in all tissues. 

### 2.5. Expression of CsADF Gene Analysis in Sweet Orange after Xcc and CLas Infection

In the expression of CsADF in sweet orange leaf after *C*Las infection, the expression of other genes except *CsADF3* was down-regulated (Figure 3). Among them, *CsADF2* and *CsADF6a* decreased significantly. Compared with the control, there was no significant difference between *CsADF3* and *CsADF5*, but the difference between *CsADF4a* and *CsADF7a* decreased by half. The decrease of *CsADF1* was smaller than that of the control. 

We also explored the expression of *CsADF* in sweet oranges at 0, 1, 4, 6, and 14 days post-infection (dpi) using *Xcc* (Figure 4). Expression of the seven *CsADF* differed. *CsADF2/4a/5/6a* genes were significantly up-regulated and peaked at 4 dpi. In particular, *CsADF4a* expression was up nearly eight-fold. Moreover, the expression of the *CsADF1/3/7a* gene reached the highest level at 6 dpi. These results suggest that the CsADFs gene may affect the resistance or susceptibility of sweet orange to *Xcc*. 

### 2.6. Subcellular Localization of CsADF Proteins

To determine the subcellular localization of CsADFs, we constructed a pCAMBIA-1132- CsADF4-GFP vector (Appendix A), and transiently expressed CsADF4 in *Nicotiana benthamiana* leaf cells using an *Agrobacterium tumefaciens* infection system (Figure 5). We found that CsADF4 was localized in both the nucleus and cytoplasm. In the control group, GFP signals were uniformly distributed in both the cytoplasm and nucleus. 

### 2.7. Transient Expression of CsADF4 Enhances Sweet Orange Susceptibility to Xcc 

To determine the role of *CsADF4* in the interaction between sweet oranges and *Xcc*, we transiently overexpressed pCAMBIA1300s-*CsADF4* and pCAMBIA1300s in sweet oranges (Appendix A). After 14 days, the symptoms of overexpressing-*CsADF4* leaves were more serious than those of the control (Figure 6A). Moreover, the diseased area of the overexpressing-*CsADF4* leaves was significantly greater than that of the control (Figure 6B). These results suggested that the expression of *CsADF4* enhanced the susceptibility of sweet oranges to *Xcc*. 

### 2.8. Analysis of CsADF4 Protein Interaction Network

Based on the PPI network database, predicted and known protein–protein interactions, including direct and indirect connections, were obtained, providing a basis for evaluating the function of uncharacterized proteins. The CsADF4 PPI network constructed in this study consisted of 11 nodes and 45 edges, with an average node degree of 8.18 (Figure 7). This network revealed the possible direct or indirect interactions between CsADF4 and other non-family members. Examples include CAP, Actin, and PP2Cc. This provides a valuable basis for further exploration of the effects of CsADF4 on the sensitivity of sweet oranges to *Xcc*. 

## 3. Discussion

*ADF* genes are highly conserved in structure and play important roles in plant development and defense. In the present study, seven *CsADF* genes were identified in sweet orange plants. Based on the phylogenetic analysis, we classified *CsADF* proteins into three subfamilies, which is consistent with previous research [5]. Collinear analysis and mapping play positive roles in revealing the close source relationships of plants [32]. Three duplicate pairs were identified. *CsADF1*, *CsADF3*, *CsADF4,* and *CsADF5* belonged to the first group; *CsADF2*, and *CsADF7* belonged to the second group; and *CsADF6* was not repeated. By estimating the ratio of non-synonymous mutations to synonymous mutations (dN/ds) of different evolutionary lineages, most of the ADF/CFL codon sites were found to be under strong purification selection, and there were few events that accelerated the evolution of proteins [10]. The three pairs of *CsADF* genes were all less than 1; they were between 0.05 and 0.06. These results showed that the ADF genes of sweet orange were selected for purification during evolution. 

Actin-depolymerizing factor is expressed in all tissues of higher plants, but its expression varies significantly [7]. Studies on *Arabidopsis* ADF have shown that subgroup I ADF has a strong structural expression in all vegetative and reproductive tissues, except pollen. Subclass II ADF is specifically expressed in mature pollen, pollen tubes, root epidermal hair mother cells, and root hairs. Subclass III ADF is weakly expressed in vegetative tissues but is most strongly expressed in fast-growing and/or differentiated cells, including callus, leaf emergence, and meristem regions, and so on [33]. AtADF1 and AtADF6 are expressed in vascular tissues of all organs, while the expression of AtADF5 was limited to the apical meristem [8]. In this study, seven *CsADF* genes were expressed in stems, leaves, and fruits. The expression of *CsADF2/4a/5/6* was the highest in fruits, whereas that of *CsADF3* was the highest in stems. ZmADF3 is specifically expressed in the apical cells of growing root hairs in maize and facilitates root hair growth by inducing actin assembly and depolymerization [14,34]. We infer that CsADF may affect the growth and development of sweet oranges. 

Actin-depolymerizing factor genes are also involved in abiotic stress [35]. For example, *Arabidopsis ADF1* participates in the salt stress response under the regulation of MYB73 [36]. DaADF3 can improve the cold tolerance of transgenic rice to adapt to extreme environments in Antarctica [37]. In this study, we analyzed the potential role of the identified *CsADF* genes in different biological processes of sweet orange by analyzing the cis-regulatory elements of the promoters and found that they contain a variety of abiotic response elements. These results suggest *CsADF* genes play important roles in abiotic stress responses. 

Therefore, the ADF gene family affects pathogenicity in higher plants [38]. In resistant Chinese cabbage varieties infected with *Botrytis cinerea*, ADF7 and ADF10 expression were up-regulated [39]. After infection with *Pseudomonas syringae tomato*, *AtADF4* was identified as necessary for the effector AvrPphB to trigger a resistance response [28]. The expression of the AvrPphB homologous resistance protein *RPS5* was significantly decreased in *adf4* mutants [29]. In the present study, we analyzed the expression of *CsADF* genes in sweet oranges infected with *Xcc* and *C*Las. *CsADF* genes showed different expression patterns. Among them, the expression of *CsADF4* reached the peak on the fourth day after infection with *Xcc* and decreased by half after infection with *C*Las. To further investigate the function of *CsADF4*, we transiently overexpressed *CsADF4* in sweet orange plants. We found the expression of *CsADF4* enhanced citrus susceptibility to *Xcc*. These results suggest that *CsADF4* plays an important role in the interaction between sweet oranges and *Xcc*. 

At present, the mechanism by which *CsADF4* enhances the sensitivity of sweet orange leaves to *Xcc* is not clear and may be affected by the regulation of upstream genes while it also may be involved in the regulation of downstream susceptible genes. According to the PPI results, there may be interactions between CsADF4 and other genes. Previous studies have shown an interaction between ADF and actin (ACT) [27]. In *Arabidopsis thaliana*, ACT7 is co-expressed with other actin isomers to promote cellular responses to external stimuli [40]. Other studies have shown that deletion of the ChCAP gene leads to a decrease in the growth rate of the conidia and hyphae of *Colletotrichum higginsianum* [41]. Therefore, it is valuable to further study its function and molecular mechanisms, combined with the predicted results of protein–protein interactions.

## 4. Materials and Methods

### 4.1. Gene Identification and Phylogenetic Analysis

Relevant sweet orange genome sequence and gene annotation information were obtained from the National Center for Biotechnology Information (NCBI) and the Sweet Orange Genome Annotation Project. All Hidden Markov Model (HMM) profile files of the ADF-H [42] domain (accession no. PF00241.23) were downloaded from the Pfam database (version 35.0; https://pfam.xfam.org/; accessed on 25 March 2022). ClustalX software v2.1 was used to perform multiple alignments of the amino acid sequences of ADF members, using default parameters and then making manual adjustments. Based on the neighbor connection (NJ) algorithm, the phylogenetic tree was generated by MEGA-X v10.2.4 software [43], including pairwise deletion, Poisson distribution, and bootstrap replicates (1000 replicates; random seed). The online software iTOL was used for beautification. 

### 4.2. Chromosome Distribution and Replication Events of CsADF Gene

The chromosomal location of the *CsADF* gene was extracted from the genome annotation information of sweet orange in the GFF3 format and visualized using the independent software TBtools v1.108 of the biologist’s toolkit [44]. Chromosome size and gene density were determined by referring to the genome annotation information of sweet oranges. *CsADF* gene replication events were identified using the multiple collinear scanning toolkit (MCScanX) program with default settings [45]. To carry out collinear analysis, the genome sequence and gene structure annotation files of sweet orange were input into one-step MCScanX, and a bilinear map plug-in was embedded in the TBtools software for visualization. TBtools toolkit v1.108 was used to calculate the Ka, Ks, and Ka/KS values. 

### 4.3. Analysis of Gene Characteristics and Structure

The theoretical isoelectronic point (pl) and molecular weight (MW) of the input proteins were estimated using TBtools v1.108. The subcellular localization of CsADF proteins was predicted using the online bioinformatics tools Plant-mPLoc (http://www.csbio.sjtu.edu.cn/bioinf/plant-multi/; accessed on 28 March 2022) and WoLF PSORT (https://wolfpsort.hgc.jp/; accessed on 28 March 2022). Based on the genome and coding sequences, the structure of each ADF gene was obtained using a Gene Structure Display server (http://gsds.cbi.pku.edu.cn/; accessed on 28 March 2022). Conserved motifs of CsADF proteins were identified using online MEME Suite Programs in classic mode. Domain-based analyses were performed using the SMART server (http://smart.embl.de/; accessed on 30 March 2022) in the default mode.

### 4.4. Cis-Acting Regulatory Elements and Protein–Protein Interaction Network Prediction

To identify CREs in the promoter sequence of sweet orange ADF genes, we extracted genomic DNA sequences extending 2000 bp upstream of the transcription start site and submitted them to the PlantCare website (http://bioinformatics.psb.ugent.be/webtools/plantcare/html/; accessed on 25 June 2022). The STRING online platform was used for protein–protein interaction (PPI) network prediction (STRING: functional protein association networks (string-db.org)). 

### 4.5. Plant Materials and Treatments

Sweet orange material for tissue-specific expression patterns and stress response analyses was obtained from the National Citrus Improvement Center of Hunan Agricultural University, Hunan Province, China. For analysis, samples of two different tissues (leaves and stems) were collected from the same sweet orange plant at the flowering stage and mature fruits were then obtained. 

The DL509 strain (Asian A line) of *Xcc* was isolated and purified at the Changsha Branch of the National Citrus Improvement Center of Hunan Agricultural University. The canker pathogen (*Xcc*), stored in the refrigerator at −80 °C, was crossed on the Luria-Bertani (LB) plate medium and cultured at 28 °C for 48 h. The activated single bacteria were picked up and cultured in LB liquid medium and shaken (200 r/min) at 28 °C overnight. After low-speed centrifugation (7500 r/min) for 5 min at 25—27 °C, the supernatant was removed, and a heavy suspension of sterile tap water was added. The OD_600_ value of the bacterial solution was determined by spectrophotometry, the concentration of the bacterial solution was adjusted, and the accurate concentration of the bacterial suspension was determined after counting with a smear plate. The bacterial solution was diluted to 10^3^, 10^4^, 10^5^, and 10^6^ cfu/mL as the injection concentration and 10^8^ cfu/mL as the acupuncture inoculation concentration.

A slightly modified version of the method described by Martins Cristina de Paula Santos et al. was used for *Candidatus* Liberibacter asiaticus (*C*Las) vaccination [46]. Sweet oranges with typical symptoms of *C*Las infection were collected from commercial citrus plantations in central and southern Hunan Province. As mentioned earlier, sweet orange sprouts without *C*Las and infected with *C*Las were grafted onto seedlings to obtain healthy and infected plants. Three independent samples were collected 48 h after vaccination and preserved as described above. The presence of *C*Las in mature leaves was detected by qRT-PCR analysis 6 months after inoculation. 

### 4.6. RNA Extraction and Real-Time Quantitative PCR

Total RNA was extracted from sweet orange leaves using TransZol (TransGen Biotech, Beijing, China), according to the manufacturer’s instructions. The Goldenstar RT6 cDNA synthesis kit (Tsingke Biotechnology, Beijing, China) was used to synthesize the first strand of complementary DNA using one microgram of total RNA. All qRT-PCR reactions were performed and analyzed using a CFX96 Touch Deep Well Real-Time PCR Detection System (Munich Bio-Rad, Munich, Germany) and SYBR Green PCR Master Mix (Beijing Solarbio, Beijing, China). qRT-PCR was performed according to standard procedures and conditions described previously. qRT-PCR gene-specific primers were designed using the Oligo7 software.

### 4.7. Subcellular Localization

The full-length amplified ADF fragment was cloned into the linearized pCAMBIAI1132 vector between the CaMV35S promoter and the green fluorescent protein tag using the ClonExpress II one-step cloning kit (Vazyme, Nanjing, China). The resulting vector was introduced into *Agrobacterium tumefaciens* EHA105 through electroporation and then infiltrated into *Nicotiana benthamiana* leaves using a needle-free syringe. Finally, the samples were observed under a Carl Zeiss LSM710 confocal laser scanning microscope. 

### 4.8. Expression Analysis of CsADF4 

The *Agrobacterium tumefaciens* EHA105 strain containing the CsADF4 plasmid and the empty vector pCAMBIA1300s (as a negative control) were resuspended in MgCl_2_ buffer to an OD_600_ of 0.6, and injected onto the functional leaves of sweet oranges that had fully expanded but not yet turned green. After 24 h of bacterial injection, the wound was slightly stabbed with a pin at the injection site, and 2 μL 10^8^ cfu/mL *Xcc* solution was dropped on the wound. After 4–7 days of inoculation, the phenotype was observed, and the area of the lesion was counted [47].

## 5. Conclusions

In this study, CsADF family proteins were analyzed comprehensively and systematically, and the effect of CsADF4 on citrus bacterial pathogens was analyzed. Seven *CsADF* genes were identified and divided into three groups that were unevenly distributed across the four chromosomes. The chromosomal position, collinearity, protein structure, conserved domain and motif, CRE, and expression profile (tissue specificity and response to citrus canker) of *CsADF* were analyzed. Significant differences were observed in the expression of CsADFs in different tissues. There were significant differences in the expression of seven *CsADF* genes in the samples infected with *Xcc*. In the samples infected with *C*Las, only five *CsADF* genes were expressed differently. This indicates that they may be involved in the immune defense response of sweet oranges. It is worth noting that overexpression of CsADF4 affected the sensitivity of sweet oranges to bacterial pathogens. These findings provide a theoretical basis for the further exploration of the roles of ADF gene family in disease resistance and susceptibility.

## Figures and Tables

**Figure 1 plants-12-03054-f001:**
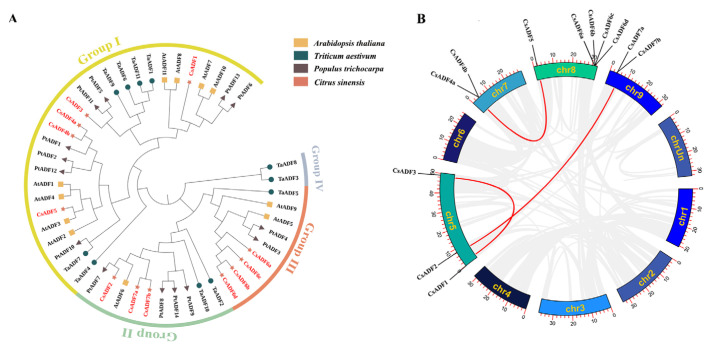
**Phylogenetic and collinear analysis of ADF family proteins.** (**A**) Phylogenetic tree of ADF proteins in *Citrus sinensis* (7 ADF proteins), *Triticum aestivum* (11 ADF proteins), *Populus trichocarpa* (14 ADF proteins), and *Arabidopsis thaliana* (11 ADF proteins). The tree shows four different phylogenetic subgroups, represented by different colors. (**B**) Circos plot displaying the collinearity of CsADF homologous genes. The connected red lines represent segmentally duplicated genes.

**Figure 2 plants-12-03054-f002:**
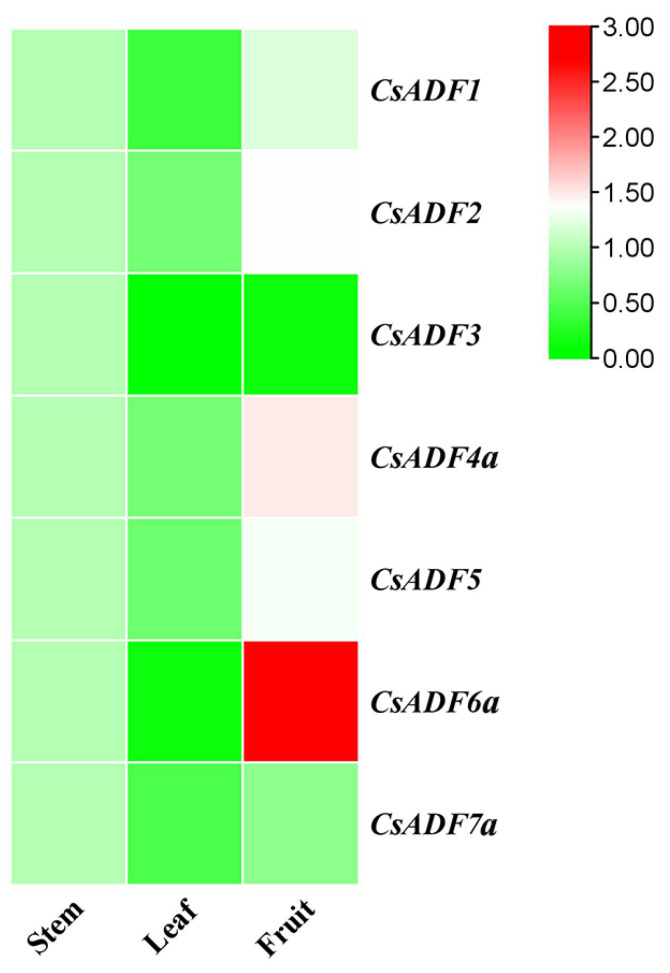
**The expression profile of *CsADF* genes among different tissues.** Based on the data collected from sweet orange plants, the transcriptional levels of *CsADF* in stems, leaves, and fruits were analyzed. The TBtools toolkit was used to generate heat maps. From green to red indicates an increase in the expression level of seven genes.

**Figure 3 plants-12-03054-f003:**
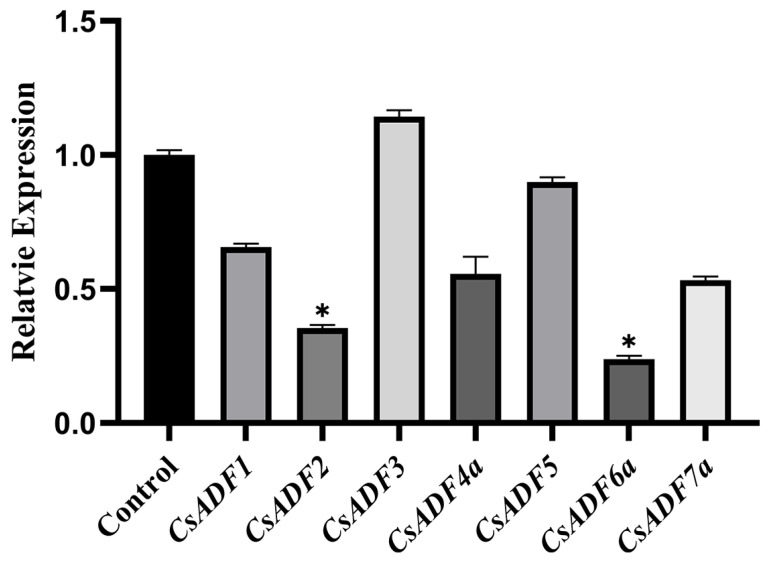
**Quantitative RT-PCR analysis of the *CsADF* gene in sweet orange leaves infected with *C*Las.** Abscissas represent gene. The ordinate represents the relative expression. The values represented the mean ± standard deviation (SD) of three independent replicates. The vertical bar represents the standard error. The ‘*’ above the horizontal line indicates a significant difference (*p* < 0.05).

**Figure 4 plants-12-03054-f004:**
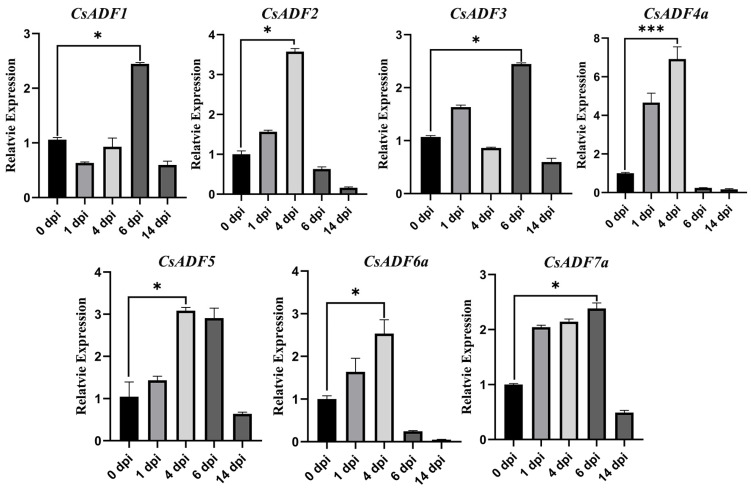
**Expression profiles of *CsADF* after infection with *Xcc* by RT-qPCR.** Abscissas denote the time point after *Xcc* vaccination. The ordinate represents the relative expression. The values represent the mean ± standard deviation (SD) of three independent replicates. The vertical bar represents the standard error. The ‘*’ above the horizontal line indicates a significant difference (*p* < 0.05). The ‘***’ above the horizontal line indicates a significant difference (*p* < 0.001).

**Figure 5 plants-12-03054-f005:**
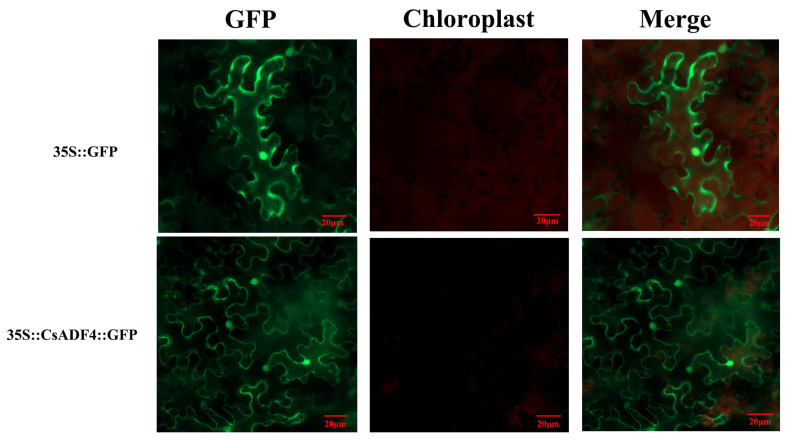
**Subcellular localization of CsADF4 in *Nicotiana benthamiana*.** With 35S::GFP as the control, three results of GFP, chloroplast, and merge are shown, respectively. Scale bars = 20 µm. 35S::GFP was used as a negative control.

**Figure 6 plants-12-03054-f006:**
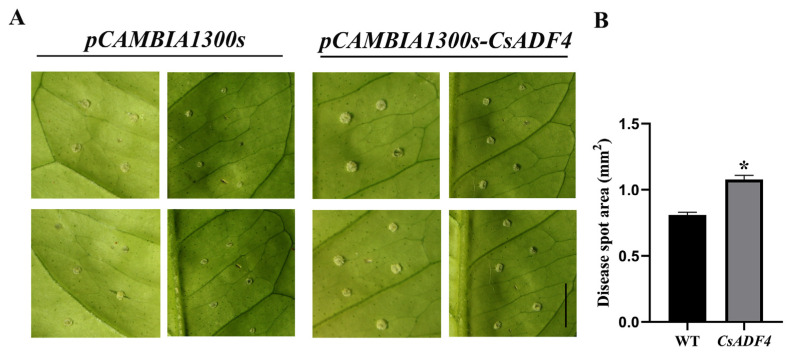
**Transient expression of *CsADF4* enhances sweet orange susceptibility to *Xcc.*** (**A**) Symptoms of *pCAMBIA1300s-CsADF4* and *pCAMBIA1300s* transient expression in leaves inoculated with *Xcc* were photographed at 14 dpi. Xcc was infiltrated at 10^8^ cfu/mL. Scale bar: 1 cm. (**B**) Statistical analysis of disease spot area. Abscissas represent genes. The ordinate represents the disease spot area. The ‘*’ above the horizontal line indicates a significant difference (*p* < 0.05).

**Figure 7 plants-12-03054-f007:**
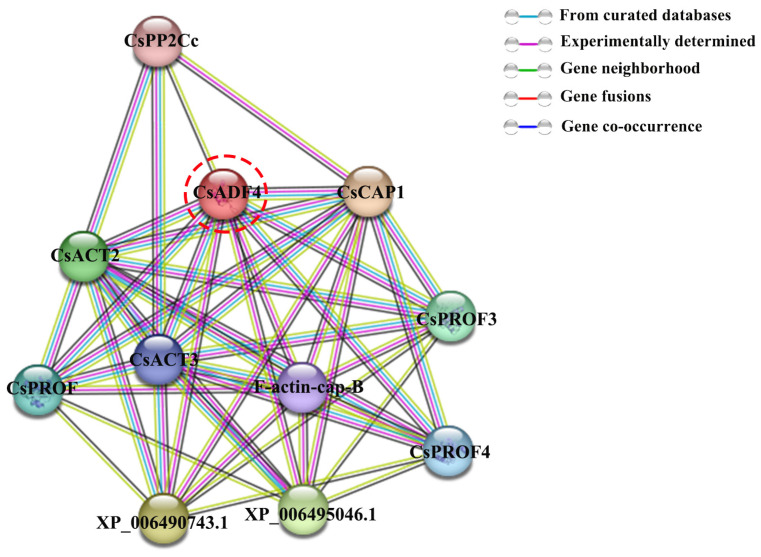
**Protein–protein interaction network prediction.** Network nodes represent and the edges represent protein–protein associations proteins. The colored nodes are query proteins and first shell of interactors. The white nodes are the second shell of interactors.

**Table 1 plants-12-03054-t001:** Information on all actin-depolymerizing factor (ADF) family genes identified in the sweet orange genome.

Gene Name	Sequence ID	Chromosomal	Protein				
			MW (kDa)	Theoretical pI	Instability Index	Aliphatic Index	GRAVY
*CsADF1*	Cs_ont_5g002950.1	chr5:1964451-1965092	16.94	8.89	43.08	74.8	−0.429
*CsADF2*	Cs_ont_5g011920.1	chr5:7702323-7704616	16.87	7.75	45.64	62.19	−0.734
*CsADF3*	Cs_ont_5g047810.1	chr5:48291662-48292851	16.06	5.29	51.16	70.86	−0.522
*CsADF4a*	Cs_ont_7g004070.1	chr7:3085584-3089118	15.96	5.31	42.83	70.86	−0.519
*CsADF4b*	Cs_ont_7g004070.2	chr7:3085584-3089118	15.96	5.31	42.83	70.86	−0.519
*CsADF5*	Cs_ont_8g003160.1	chr8:1684426-1686707	19.73	6.31	50.57	67.95	−0.429
*CsADF6a*	Cs_ont_8g027730.1	chr8:31643742-31645476	16.32	6.73	23.24	70.28	−0.214
*CsADF6b*	Cs_ont_8g027730.2	chr8:31643742-31645476	13.25	6.08	13.1	80.52	0.148
*CsADF6c*	Cs_ont_8g027730.3	chr8:31643742-31645476	13.25	6.08	13.1	80.52	0.148
*CsADF6d*	Cs_ont_8g027730.4	chr8:31643742-31645476	13.25	6.08	13.1	80.52	0.148
*CsADF7a*	Cs_ont_9g008180.1	chr9:5690959-5694031	16.58	6.84	34.24	65.45	−0.669
*CsADF7b*	Cs_ont_9g008180.2	chr9:5690959-5694031	16.58	6.84	34.24	65.45	−0.669

## Data Availability

All available data are reported in the paper.

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
