# Peer review of "Actin-Depolymerizing Factor Gene Family Analysis Revealed That CsADF4 Increased the Sensitivity of Sweet Orange to Bacterial Pathogens"

_plants, 2023, doi:10.3390/plants12173054_

Round 1

Reviewer 1 Report

I consider that the work is limited to describing the results without functionally contextualizing their structural and expression characteristics in relation to their possible participation in tolerance or susceptibility to pathogens.

2.1 Genome-wide identification and chromosome mapping of the ADF gene in sweet orange.

The information they present in relation to the identification and characterization of the ADF genes in Citrus is very complete and clear. In addition, its information is complemented with the position in the different chromosomes. The phylogenetic tree outlines a generation scheme for these genes during evolution and outlines the possible sequences associated with the different functions that these genes have in plants.

I would make the following comments:

2.3 Structural and cis-acting regulatory element analysis of CsADF genes

I consider it important to highlight the differences in the type of domains that the ADF genes have and their possible functional significance.

2.4 Expression profile of CsADF genes in different tissues of sweet orange

In figure 2, the different expression level of CsADF3 compared with the rest CsADFs, is not clear as they mention in the text.

It is important to mention which could be the real meaning of the high expression of CsADF6a in fruits.

2.5. Expression of CsADF gene analysis in sweet orange after Xcc and CLas infection

CsADF6a has a very high expression in fruits and is significantly reduced in CLas-infected tissue. It would be interesting to refer if this is associated with the characteristics of resistance and susceptibility.

I consider it important to relate the maximum expression time of the CsADFs genes with the resistance and/or susceptibility characteristics.

2.7. Transient expression of CsADF4 enhances sweet orange susceptibility to Xcc

Why is the analysis done at 14 dpi and not at 4 dpi when this gene has its maximum expression?

Author Response

We would like to express our gratitude for providing us with the opportunity to improve the quality of our submitted manuscript titled "Actin Depolymerizing Factor gene family analysis revealed that CsADF4 increased the sensitivity of sweet orange to bacterial pathogens." We sincerely appreciate the constructive and insightful comments provided by the reviewers. In this revised version, we have diligently addressed all of the comments and suggestions made. We firmly believe that the revised manuscript now meets the publication standards of the journal.

To make it easier for you to identify the revisions, we have highlighted them in red throughout the manuscript.

In the following pages, we have provided point-to-point responses to the queries raised by the reviewers.

Reviewer#1

Comment 1 (2.3 Structural and cis-acting regulatory element analysis of CsADF genes): " I consider it important to highlight the differences in the type of domains that the ADF genes have and their possible functional significance."

Response: Thank you very much for your suggestions on this manuscript. According to your suggestion, we have added a description of the domain type and function. Change “We identified five conserved motifs in CsADFs, designated as motifs 1 to 5, and each CsADF had five conserved motifs, except for CsADF6b/c/d, which had four. All CsADF proteins contained ADF domains.” to “ We identified five conserved motifs in CsADFs, designated as motifs 1 to 5, and each CsADF had five conserved motifs, except for CsADF6b/c/d, which had four. All CsADF proteins contained ADF domains. CsADF1/2/3/6b/6c/6d/7a/7b contains ADF-gelsolin supe domains, which increase the turnover rate of actin and interact with actin monomers and actin filaments. CsADF4a/4b/5 contains ADF-cofilin-like domains, which enhance the turnover rate of actin, and interact with actin monomers (G-actin) as well as actin filaments (F-actin).” (Line 137-143).

Comment 2 (2.4 Expression profile of CsADF genes in different tissues of sweet orange): "In figure 2, the different expression level of CsADF3 compared with the rest CsADFs, is not clear as they mention in the text. It is important to mention which could be the real meaning of the high expression of CsADF6a in fruits. "

Response:

Thank you very much for your suggestions on this section, which have made its expression clearer. We have made the necessary changes and highlighted them in red (Line 159-162).

Comment 3 (2.5. Expression of CsADF gene analysis in sweet orange after Xcc and CLas infection): "CsADF6a has a very high expression in fruits and is significantly reduced in CLas-infected tissue. It would be interesting to refer if this is associated with the characteristics of resistance and susceptibility. "

Response: Thank you very much for your suggestion. The tissue infected with Clas in this article is sweet orange leaves rather than fruits, so we did not compare them. Nevertheless, we will seriously consider your suggestion and explore whether it can be linked together in subsequent experiments. Once again, we appreciate your thoughtful advice and attention. If you have any other questions or suggestions, we would be more than happy to listen. Thank you!

Comment 4 (2.5. Expression of CsADF gene analysis in sweet orange after Xcc and CLas infection): "I consider it important to relate the maximum expression time of the CsADFs genes with the resistance and/or susceptibility characteristics. "

Response: Thank you for your suggestion. The time shown in the results is the days after receiving the disease, not the days of overexpression of the gene. Because the maximum expression time of Huanglong disease is not stable, we take the diseased leaves at a specific time for analysis. According to your suggestion, we revised the original text and highlighted it in red. (Line 183-184).

Comment 5 (2.7. Transient expression of CsADF4 enhances sweet orange susceptibility to Xcc): "Why is the analysis done at 14 dpi and not at 4 dpi when this gene has its maximum expression?”

Response: Thank you for your professional review. On the fourth day, the leaves after inoculation already had different phenotypes visible, but the lesion area was small and it was not easy to count, so we combined with the methods of others' studies and chose the 14th day to display [1].

Reviewer#2

Comment 1: “The manuscript needs an extensive english revivion before publication.”

Response: Thank you very much for your valuable advice. We sincerely thank you for your feedback and carefully review and revise the article.

In addition, we have made the following changes to the manuscript:

  1. For the questions raised by the editor: “We noticed that there are duplicate references between 10 and 11, 15 and 38, and 8 and 36. Please carefully check and revise or remove the duplicate one while revision.”

Response: Thank you very much for your valuable feedback. We appreciate your comments and carefully reviewed the manuscript, changing the citations "11" to "10" and "36" to "8" marked in red (Line 58, Line 249). Citations "15 and 38" do not belong to the same article.

  1. Change “Alexandra Gentile” to “Alessandra Gentile” and mark it in red (Line 5).
  2. Change “2National Citrus Improvement Center of Hunan Agricultural University (Changsha Branch) Changsha 410128, China; College of Horticulture, Hunan Agricultural University Changsha 410128, China, dsm531@126.com (S.D.); ldazhi@163.com (D.L.); suyajun2565@163.com (Y.S.)” to “2 National Citrus Improvement Center of Hunan Agricultural University (Changsha Branch) Changsha 410128, China; (S.D.); ldazhi@163.com (D.L.); suyajun2565@163.com (Y.S.); gentilea@unict.it (AG); 15934961987@163.com (J.X.); wang20220908@163.com (X.W.); zhufu@hunau.edu.cn (B.W.) 3College of Horticulture, Hunan Agricultural University Changsha 410128, China, dsm531@126.com (S.D.); ldazhi@163.com (D.L.); suyajun2565@163.com (Y.S.) (Line 9-13).
  3. Change “Jing Xu1,2†, Suming Dai2†, Xue Wang1,2, Alexandra Gentile3, Zhuo Zhang4, Qingxiang Xie1, Yajun Su2, Dazhi Li2* and Bing Wang1,2*” to “Jing Xu1,2†, Suming Dai2,3†, Xue Wang1,2, Alessandra Gentile2,4, Zhuo Zhang5, Qingxiang Xie1, Yajun Su2,3, Dazhi Li2,3* and Bing Wang1,2*”(Line 5-6).
  4. Change “After inoculation, the phenotype was observed, and the area of the lesion was counted.” to “After 4-7 days of inoculation, the phenotype was observed, and the area of the lesion was counted.” Reference 50 has been added. (Line 375-376).

Once again, we express our gratitude for your valuable feedback and guidance throughout the review process. We are confident that the improvements made will strengthen the scientific content and clarity of the paper. We hope the revised manuscript meets your expectations and look forward to your further evaluation.

Best regards,

College of Horticulture,

Hunan Agricultural University,

Changsha, Hunan 410128, P.R. China.

E-mail address: zhufu@hunau.edu.cn (B.W.); ldazhi@163.com (D.L.)

References

  1. Du J; Wang, Q.; Shi, H.; etc. A prophage-encoded effector from "Candidatus Liberibacter asiaticus" targets ASCORBATE PEROXIDASE6 in citrus to facilitate bacterial infection. Mol Plant Pathol 2023, 24, 302-316, doi:10.1111/mpp.13296.

Reviewer 2 Report

The manuscript highlights new aspects on the role of actin-depolymerizing factor (ADF) gene family in the sweet orange Citrus sinensis, in terms of gene sistematic identification and their expressions in different organs and under biotic stress. The aim of the study is clearly presented and is supported by literature cited in the introduction. The methods used are well presented and trustable, and coclusions are well supported by the results showed. The manuscript needs an extensive english revivion before publication.

The manuscript needs an extensive english revision.

Author Response

(The authors gave the same response as above.)
